# HPLC-Based Purification and Isolation of Potent Anti-HIV and Latency Reversing Daphnane Diterpenes from the Medicinal Plant *Gnidia sericocephala* (*Thymelaeaceae*)

**DOI:** 10.3390/v14071437

**Published:** 2022-06-30

**Authors:** Babalwa Tembeni, Amanda Sciorillo, Luke Invernizzi, Thomas Klimkait, Lorena Urda, Phanankosi Moyo, Dashnie Naidoo-Maharaj, Nathan Levitties, Kwasi Gyampoh, Guorui Zu, Zhe Yuan, Karam Mounzer, Siphathimandla Nkabinde, Magugu Nkabinde, Nceba Gqaleni, Ian Tietjen, Luis J. Montaner, Vinesh Maharaj

**Affiliations:** 1Department of Chemistry, University of Pretoria, Pretoria 0028, South Africa; btembeni2@gmail.com (B.T.); lukeinvernizzi@gmail.com (L.I.); phanankosimoyo@gmail.com (P.M.); dashnie.naidoo@up.ac.za (D.N.-M.); 2The Wistar Institute, Philadelphia, PA 19104, USA; asciorillo@wistar.org (A.S.); nlevitties@gmail.com (N.L.); kgyampoh@wistar.org (K.G.); gzu@wistar.org (G.Z.); zyuan@wistar.org (Z.Y.); itietjen@wistar.org (I.T.); 3Molecular Virology, Department of Biomedicine, University of Basel, 4031 Basel, Switzerland; thomas.klimkait@unibas.ch (T.K.); lorena.urda@unibas.ch (L.U.); 4Agricultural Research Council-Vegetables, Industrial and Medicinal Plants, Private Bag X293, Pretoria 0001, South Africa; 5Jonathan Lax Immune Disorders Treatment Center, Philadelphia Fight Community Health Centers, Philadelphia, PA 19107, USA; mounzerk@fight.org; 6Ungangezulu Indigenous Remedies, J Uitval, Wasbank 2920, South Africa; nceba5850@gmail.com (S.N.); magugun@webmail.co.za (M.N.); 7Africa Health Research Institute, Congella 4013, South Africa; nceba.gqaleni@ahri.org; 8Discipline of Traditional Medicine, University of KwaZulu-Natal, Durban 4001, South Africa; 9Faculty of Health Sciences, Durban University of Technology, Durban 4001, South Africa

**Keywords:** antiretroviral therapy, HIV, *Gnidia sericocephala*, daphnane-type compounds, reverse HIV latency, protein kinase C activation, HIV in PBMC

## Abstract

Despite the success of combination antiretroviral therapy (cART), HIV persists in low- and middle-income countries (LMIC) due to emerging drug resistance and insufficient drug accessibility. Furthermore, cART does not target latently-infected CD4+ T cells, which represent a major barrier to HIV eradication. The “shock and kill” therapeutic approach aims to reactivate provirus expression in latently-infected cells in the presence of cART and target virus-expressing cells for elimination. An attractive therapeutic prototype in LMICs would therefore be capable of simultaneously inhibiting viral replication and inducing latency reversal. Here we report that *Gnidia sericocephala*, which is used by traditional health practitioners in South Africa for HIV/AIDS management to supplement cART, contains at least four daphnane-type compounds (yuanhuacine A (**1**), yuanhuacine as part of a mixture (**2**), yuanhuajine (**3**), and gniditrin (**4**)) that inhibit viral replication and/or reverse HIV latency. For example, **1** and **2** inhibit HIV replication in peripheral blood mononuclear cells (PBMC) by >80% at 0.08 µg/mL, while **1** further inhibits a subtype C virus in PBMC with a half-maximal effective concentration (EC_50_) of 0.03 µM without cytotoxicity. Both **1** and **2** also reverse HIV latency in vitro consistent with protein kinase C activation but at 16.7-fold lower concentrations than the control prostratin. Both **1** and **2** also reverse latency in primary CD4+ T cells from cART-suppressed donors with HIV similar to prostratin but at 6.7-fold lower concentrations. These results highlight *G. sericocephala* and components **1** and **2** as anti-HIV agents for improving cART efficacy and supporting HIV cure efforts in resource-limited regions.

## 1. Introduction

Despite the resounding success of combination antiretroviral therapy (cART), HIV/AIDS remains a global public health threat, with 37.7 million people living with the virus as of 2020 [1]. Moreover, the emergence of drug resistant HIV strains threatens to reverse the historical gains made to curb this epidemic [2]. HIV drug resistance is disproportionately found in low-to-middle income countries (LMICs) [3]. For example, in South Africa, pretreatment drug resistance to non-nucleoside reverse transcriptase inhibitors (PDR NNRTI) has already surpassed the WHO-recommended 10% threshold and has necessitated a change to dolutegravir-based HIV first line therapy [4,5]. As of 2021, approximately, 7.9 million people in South Africa are living with HIV, with only 4.7 million people having access to HIV treatment, and inequalities in HIV treatment are a contributing factor to the spike in new infections [6]. As a result, new anti-HIV agents continue to be needed to address emerging drug resistance and to improve cART accessibility, particularly in LMICs and other resource-constrained regions.

An additional barrier to current HIV management efforts is the presence of latently infected CD4+ T cells, where HIV-1 provirus has incorporated into the host genome and where it escapes clearance by cART and detection by the immune system [7,8]. As these latently-infected cells can reactivate at any time to produce new infectious virus, individuals with HIV infection require continuous cART administration for life [9]. One experimental method to identify and eliminate latently-infected CD4+ T-cells, frequently termed “shock-and-kill,” involves the use of latency reversal agents (LRAs) to induce viral expression in latency-infected cells in the co-presence of immunotherapy support and cART [10]. Reactivated cells, in principle, should then be eliminated through host immune control, while the co-administration of cART should prevent viral spread and the re-seeding of the viral reservoir [11]. However, while several LRAs have been investigated in clinical trials, no LRA to date has consistently reduced the viral reservoir in humans [12,13]. As a result, new agents with an improved ability to reverse HIV latency and target affected cells for elimination are still required.

Toward improving drug-resistant HIV management and cure efforts in LMICs, one attractive therapeutic prototype would be a small molecule that can simultaneously inhibit viral replication and induce latency reversal. In the realm of natural product-based chemical compounds, two agents, including prostratin, originally isolated from the tree *Homalanthus nutans* (*Euphorbeaeceae*), and bryostatin, a macrolide lactone originally isolated from the marine organism *Bugula neritina*, establish a precedent for these prototypes [12,14]. For example, both prostratin and bryostatin can inhibit HIV-1 infection by downregulating the CD4 entry receptor of HIV along with HIV co-receptors CCR5 and CXCR4, thereby antagonizing the ability of HIV to gain entry into uninfected T cells [15,16]. Prostratin and bryostatin additionally reverse HIV latency by activating protein kinase C and downstream NF-kB signaling, thereby inducing viral transcription at the integrated HIV promoter [14]. However, as the limited activity of prostratin and the limited availability of bryostatin preclude their immediate development in many LMICs, additional highly-active and dual-acting natural products that are readily accessible are required.

To identify new, natural product-based agents that both inhibit HIV replication and reverse HIV latency from local sources, we investigated the plant species *Gnidia sericocephala* (Thymelaeaceae). This plant species was selected based on its use by traditional health practitioners for HIV/AIDS management in the KwaZulu Natal province of South Africa (S. Nkabinde and M. Nkabinde, personal communication). Using a high-performance liquid chromatography (HPLC)-based profiling approach, we demonstrate that the identified compounds, yuanhuacine A and a mixture containing yuanhuacine, can inhibit HIV replication and/or reverse HIV latency in vitro and ex vivo.

## 2. Materials and Methods

### 2.1. Sequential Extraction of Gnidia sericocephala Roots

*Gnidia sericocephala* roots were collected from Tugela Ferry in the Msinga region of KwaZulu Natal, South Africa. The identity of the collected plant species was confirmed by a curator at the H.G.W.J Schweickerdt Plant Herbarium at the University of Pretoria where a voucher specimen (No. 126590) was deposited. Dried ground roots of *G. sericocephala* were sequentially extracted using hexane, dichloromethane (DCM), ethyl acetate (EtOAc), and methanol (MeOH). Extracts were filtered under vacuum using Whatman No. 1 filter paper. All extracts were evaporated until dry using a rotary evaporator and stored at 4 °C prior to biological screening and analysis.

### 2.2. Microfractionation of the DCM Extract

Microfractionation of the DCM extract was performed on a Waters chromatographic system (Waters 600 controller, Waters Corporation, Milford, MA, USA). All separation was done on a Luna Phenomenex reverse phase column (250 × 4.60 mm, 5 μm). The mobile phase consisted of H_2_O (0.05% trifluoroacetic acid) as solvent A and acetonitrile (ACN) (0.05% trifluoroacetic acid) as solvent B. A gradient elution of 5–95% solvent B was carried out for 30 min at a flow rate of 1 mL/min. An injection volume of 30 µL corresponding to 300 µg of sample was used. Data were collected using Millennium^®^ software (Waters Corporation, Milford, MA, USA). A total of 30 fractions were collected in a 96 deep-well plate, corresponding to approximately 10 µg of the fractions in each well of the plate. This was repeated three times before evaporating until dry in a Genevac EZ-2 centrifugal evaporator (Genevac Ltd., Ipswich, UK). The 30 microfractions were dissolved in dimethyl sulfoxide (DMSO) and subjected to anti-HIV screening using the deCIPhR assay as previously described [17]. Five bio-active microfractions were combined and subjected to UPLC-QTOF-MS/MS analysis to tentatively identify the compounds in the fractions.

### 2.3. UPLC-QTOF-MS/MS Characterization of Microfractions

The combined bio-active microfractions were chemically characterized using a Waters Acquity UPLC system (Waters Corporation, Milford, MA, USA) equipped with a binary solvent delivery system. The auto sampler was operated using MassLynx v4.1 software (Waters Corporation, Milford, MA, USA). MS calibration was performed using sodium formate clusters over a mass range of 50–1200 Da to achieve an absolute mass accuracy of <0.5 mDa. Separation of compounds was achieved using a Waters BEH C18 reverse phase column (2.1 mm × 100 mm, 1.7 μm). The mobile phase consisted of H_2_O (0.1% formic acid) as solvent A and ACN (0.1% formic acid) as solvent B. A gradient elution method was used and ran as follows: an initial hold at 3% solvent B (0–0.1 min) before a linear increase to 100% solvent B (0.1–14 min), and an isocratic hold at 100% solvent B (14–16 min) before reconditioning the column with the starting conditions (16.50–20.00 min). The flow rate was maintained at 0.4 mL/min with an injection volume of 5 µL. The column temperature was kept constant at 40 °C. The following MS source parameters were set for both positive and negative electrospray modes: source temperature of 100 °C, sampling cone voltage of 15 V, extraction cone voltage of 4.0 V, desolvation temperature of 400 °C, cone gas flow of 10.0 L/h, desolvation gas flow of 700 L/h, and a capillary voltage of 2.0 kV. Sample analysis was conducted in both positive and negative ionisation modes; however, due to the nature of the compounds, positive ionisation mode was selected for further analyses.

### 2.4. Isolation and Purification of Compounds from the DCM Extract

Isolation of yuanhuacine A (**1**) and yuanhuacine (as a mixture containing an additional isomer) (**2**): The DCM extract (16 g) was initially separated using open column chromatography employing silica gel as the stationary phase and the following solvent systems as mobile phases: DCM:n-Hexane (1:1), acetone, and acetone:MeOH (4:1) as a column wash at a flow rate of 5 mL/min. Collected fractions were pooled based on their thin layer chromatographic profile to yield 13 fractions. All fractions were subjected to UPLC-QTOF-MS/MS analysis. Fraction BT22-6/11 (250 mg) showed a similar UPLC-QTOF-MS/MS chemical profile consistent with that of the 5-pooled bio-active microfractions. The fraction was dissolved in ACN (100%) and filtered through a 0.22 μm filter to remove any particles and subjected to further purification using preparative high-performance liquid chromatography (prep HPLC). A Waters chromatographic system was used, fitted with a Waters PDA (Model 2998), and interfaced with an ACQUITY QDa detector (Waters Corporation, Milford, MA, USA).

The isolation of compound **1** was carried out on an Xbridge^®^ preparative C18 OBD™ (19 mm × 250 mm, 5 μm) column where solvent A consisted of H_2_O (0.1% formic acid) and solvent B consisted of ACN (0.1% formic acid). The flowrate was kept constant at 20 mL/min. A gradient elution method was employed which ran as follows: an initial hold at 5% solvent B (0–1 min), followed by a linear increase to 50% solvent B (1–8 min), and an additional linear increase to 85% solvent B (8–13 min) before returning to the starting conditions (13–20 min). An injection volume of 150 μL was used.

Additionally, the isomeric mixture containing compound **2** was obtained using the same system as above, where separation was conducted on an Xbridge^®^ preparative C18 OBD™ (10 mm × 250 mm, 5 μm) column where solvent A consisted of H_2_O (0.1% formic acid) and solvent B consisted of ACN (0.1% formic acid). A fixed flowrate of 5 mL/min was used with the elution method optimised as follows: an initial hold at 70% solvent B (0–1 min) before a linear increase to 100% solvent B (1–13 min) before an isocratic hold at 100% solvent B for 2 min (13–15 min) before reverting to the starting conditions (15–15.5 min) and re-equilibrating the column (15.5–19.5 min). The UV scan range was set at 210–400 nm.

Data were collected using MassLynx v4.1 (Waters Corporation, Milford, MA, USA) software. Positive ionization mode was selected for data collection. The probe temperature and source temperature were set at 600 °C and 120 °C, respectively. The capillary and cone voltages were set to 1.5 kV and 10 V, respectively. Data were collected in a continuous form with mass to charge ratios (*m*/*z*) of 300 to 1000 Da recorded. The pure compound yuanhuacine A (3.1 mg) was collected based on mass (*m/z* 621), similarly with the yuanhuacine mixture (*m/z* 648) (2.4 mg) using a fraction collector and dried using a Genevac EZ-2.

Isolation of yuanhuajine (**3**) and gniditrin (**4**): Further fractionation of the DCM extract was carried out using flash chromatography on a BUCHI Pure C-815-Flash system (Buchi, Flawil, Switzerland) fitted with a UV detector and an ELSD. The extract (20 g) was injected onto a Buchi EcoFlex silica cartridge (220 g, 50 µm). The sample was eluted using a hexane (A) and acetone (B) gradient mixture and was optimised as follows: the starting conditions were a linear increase to 15% solvent B over 5 min (0–5 min), followed by an increase to 20% solvent B over 20 min (5–25 min), and another linear increase to 30% solvent B over 10 min (25–35 min); before reaching 100% solvent B, a linear increase to 50% solvent B over 10 min (35–45 min) was applied, and finally with an increase to 100% solvent B over 20 min (45–65 min) at a flowrate of 120 mL/min over 80 min. Compounds were detected using the ELSD and UV at wavelengths (215, 230, 254, and 279 nm). Fractions were chemically profiled using UPLC-QTOF-MS/MS. One fraction BT-8 (150 mg) was observed to contain daphnane diterpenoids based on its UPLC-QTOF-MS/MS fingerprint. Two compounds, yuanhuajine (**3**) and gniditrin (**4**), both corresponding to *m/z* 647 (3.1 and 0.89 mg) were collected using mass directed purification as was carried out for yuanhuacine A and yuanhuacine above. Structure elucidation of compounds **1**–**4** was carried out using data obtained from UPLC-QTOF-MS/MS, and 1D and 2D NMR. The ^1^H, ^13^C, and 2D NMR spectral data were acquired on a Bruker Avance III HD 500 MHz NMR spectrophotometer at 25 °C with Prodigy Probe. The compounds were dissolved in deuterated chloroform (CDCl_3_) (Aldrich Chemistry, Sigma-Aldrich, Milwaukee, WI, USA) and the Chemical shifts reported in ppm, referenced to residual solvent resonances (CDCl_3_ δ_H_ 7.26, δ_C_77.16 ppm) [18].

### 2.5. Anti-HIV and Cytotoxicity In Vitro Screening of Compounds and Extracts of Gnidia sericocephala

The hexane, DCM, EtOAc, and MeOH extracts, together with the compounds **1**, **2** (as an isomeric mixture), **3,** and **4** were dissolved in DMSO to create a stock solution of 10 mg/mL. These were subjected to anti-HIV (NL4-3 isolate) in vitro screens using a dual enhancement of cell infection to phenotypic resistance assay (deCIPhR) assay [2,17]. Dose-response investigations were carried out for all extracts at a 5-point concentration ranging from 200 µg/mL to 0.8 µg/mL (3-fold serial dilution in triplicate). Efavirenz and DMSO served as positive and negative inhibitor controls in the assay, respectively. Cytotoxicity was assessed under the same cell culture conditions used for the anti-HIV activity. After four days of incubation, cytotoxicity was determined using Alamar Blue reagent [17].

### 2.6. Anti-HIV Screening of Compound 1 in PBMC

Compound **1** was screened ex vivo for its ability to inhibit HIV-1 (PBL286 (02ET_14), subtype C, Ethiopian isolate) replication in fresh human peripheral blood mononuclear cells (PBMC) (seronegative for HIV and HBV). Assessment of compound efficacy was carried out by the Southern Research Institute (SRI), (Project #16174.010), Maryland, USA, as previously described [19,20]. Full-dose assays, with a 9-point concentration ranging from 10 µg/mL to 0.001 µg/mL (half-log dilution scheme, in triplicate), were carried out for anti-HIV activity and PBMC viability with azidothymidine (AZT) serving as a positive control [19,20]. EC_50_ (half-maximal effective concentration) and CC_50_ (half-maximal cytotoxic concentration) values were calculated using an SRI in-house computer program. Results are reported as mean ± SD conducted in 3 technical repeats.

### 2.7. In Vitro Latency Reversal, Cell Viability, and T Cell Activation Assays

Compounds **1** and **2** (as an isomeric mixture) were investigated for their ability to reverse HIV latency using J-Lat T cells (clone 10.6) obtained from the NIH AIDS Reagent Program, Division of AIDS, NIAID, NIH (contributed by Dr. Eric Verdin) [21]. Cells were cultured in R10+ media [RPMI 1640 with HEPES and L-Glutamine, 10% Fetal Bovine Serum (FBS), 100 U of penicillin/mL, and 100 μg of streptomycin/mL (Sigma-Aldrich, St. Louis, MO, USA) and incubated at 37 °C and 5% CO_2_. Prostratin and Gö 6983 were purchased from Sigma-Aldrich. Enzastaurin was purchased from Sellick Chemicals GmbH through ThermoFisher (Waltham, MA, USA).

Cells were seeded into 96-well plates at 2 × 10^5^ cells/well in the presence of test agents at defined concentrations or 0.1% DMSO vehicle control and incubated as described above for 24 h. GFP expression in J-Lat 10.6 cells was monitored by flow cytometry using a FACSCelesta analyzer (BD Biosciences, Franklin Lakes, NJ, USA), and data were analyzed using FlowJo v. 10.5.3 (FlowJo LLC; Ashland, OR, USA). All presented results represent the mean ± S.D. from 3 independent experiments. To measure cell viability, Jurkat T cells (Clone E6-1) obtained from the American Tissue Culture Collection (Manassas, VA, USA) were treated as described above before the addition of resazurin (Sigma-Aldrich) to a final concentration of 20 μg/mL. Cells were then incubated for an additional 4 h before fluorescence intensity was measured using a ClarioStar plate reader (BMG Labtech). Background fluorescence was subtracted from wells containing resazurin and R10+ medium but no cells.

To measure T cell activation, Jurkat cells were treated as described above before staining with CD69-phytoerythrin (BD Biosciences; Franklin Lakes, NJ, USA) antibody using the Cytofix/Cytoperm fixation/permeabilization kit (BD Biosciences) according to the manufacturer’s instructions. Flow cytometry was then performed as described above.

### 2.8. Ex Vivo Latency Reversal Assays

Compounds **1** and **2** (as an isomeric mixture) were tested for ability to reverse HIV latency using primary cell isolates. Primary CD4+ T cells were obtained from three HIV-1 infected donors on stably suppressive cART (< 50 copies/mL of plasma viral load) for at least 3 years, following informed written consent. These participants were recruited in accordance with human subject research guidelines of the United States Department of Health and Human Services under the supervision of the Wistar and Philadelphia FIGHT institutional review boards. CD4+ T cells were isolated from PBMC using the EasySep Human CD4+ T cell Enrichment Kit (STEM-CELL, Vancouver, VBC, Canada) and cultured in R20+ media, which was identical to R10+ media described above except for addition of 20% FBS.

Primary cell viability and latency reversal in the presence of test agents were performed as described previously [22]. Briefly, 2 × 10^6^ cells were treated with 0.1% DMSO, test agent, or anti-CD3/CD28 dynabeads (Invitrogen, Carlsbad, CA, USA) in a 1:1 ratio for 24 h. Following incubation, live cells were counted visually by trypan blue staining. Viral RNA was extracted from culture supernatants using a QIAmp Viral RNA Mini kit (Qiagen, Germantown, MD, USA) and subjected to quantitative RT-PCR as described previously using a C1000 Thermal Cycler and CFX96 Real-Time system (Bio-Rad, Hercules, CA, USA) [23]. The limit of detection of supernatant viral RNA was 10 copies/mL as determined through repeating end point detection from serial dilution of the AcroMetrix HIV-1 Panel (ThermoFisher).

## 3. Results and Discussion

### 3.1. Anti-HIV Activity of G. Sericocephala Extracts

The four extracts obtained by the sequential extraction of *G. sericocephala* roots were screened for their activity to block HIV replication in vitro in a dose-response manner using the deCIPhR assay [17]. Briefly, the HeLa-SXR5 cell line was infected with the CXCR4-tropic HIV isolate NL4-3 [24], and the infection was monitored in SXR5 cells due to the β-galactosidase expression of an HIV long terminal repeat (LTR)-driven LacZ gene. A reduced or eliminated β-galactosidase activity in the infected cells in the presence of the test agent thus indicated the inhibition of HIV replication. Using this assay, efavirenz served as a positive control with a calculated half-maximal effective concentration (EC_50_) of 598.4 nM, consistent with previous studies (Appendix A) [17]. When tested at 0.8 μg/mL, the dichloromethane extract emerged as the most active sample with 100% inhibition of HIV replication (Figure 1), while demonstrating no overt cytotoxicity (Appendix A). This extract was then selected for microfractionation.

### 3.2. Anti-HIV Activity of HPLC Microfractions of the DCM Extract

Having shown the most pronounced activity, the antiviral activity of the DCM extract was localised by means of HPLC-based activity on a semi-preparative HPLC. The profile of the combined microfractions that were analysed by UPLC-QTOF-MS/MS and the corresponding anti-HIV activity evaluated at 2.5 µg/mL in a deCIPhR assay are shown in Figure 2. The microfractions 25, 26, 27, 28, and 29 inhibited the in vitro virus replication by 52.0 ± 0.9%, 71.0 ± 5.9%, 83.9 ± 9.6%, 103 ± 2.1%, 101 ± 2.0%, and 91.6 ± 2.8%, respectively. The UPLC-QTOF-MS/MS profile of the combined microfractions showed the presence of coumarins and daphnane diterpenoids. The daphnane diterpenoids corresponding to the *m*/*z* 621.2669 (peak 1), *m*/*z* 647.2833 (peak 2 and 3), and *m*/*z* 649.3020 (peak 4) were tentatively identified. As these compounds are known for potent anti-HIV properties via down regulating the chemokine receptor (CXCR4) used by the virus in cell entry [17,25], they were hypothesised to be responsible for the observed anti-HIV activity in the microfractions and were prioritised for isolation and purification. Four daphnane diterpenoid components were subsequently isolated and characterised using NMR spectroscopy.

### 3.3. NMR and MS Analysis of Purified Compounds

Compound **1**: Compound **1** was identified as yuanhuacine A and obtained as a pale-yellow amorphous powder. Based on the UPLC-QTOF-MS/MS analysis, under ESI positive ionisation mode, yuanhuacine A appeared at *m/z* 621.0960 [M+H]^+^ with a corresponding molecular formula of C_35_H_40_O_10_ (calculated 621.0360) with 16 degrees of unsaturation, correlating to that observed in the literature [26]. The ^1^H-NMR spectrum revealed proton signals and ^13^C atoms which were in accordance with the molecular formula. The ^1^H-NMR spectrum indicated the presence of four methyl groups at δ_H_ 1.89 (3H, br. s, H-17), 1.79 (3H, dd, *J* = 1.27 Hz, H-19), 1.41 (3H, d, *J* = 7.69 Hz, H-18), and 0.93 (3H, t, *J* = 7.3 Hz, H-8′). The presence of a phenyl group was shown by the signals at δ_H_ 7.92 (2H, dd, *J* = 8.2, 8.4 Hz, C-3″ and C-7″), 7.48 (2H, t, *J* = 8.1, 15.2 Hz, C-4″ and C-6″), and 7.61 (1H, m, C-5″) (Table 1). The DEPT NMR spectrum showed the presence of an α,β-unsaturated ketone at δ_C_ 209.50 (C-3). DEPT confirmed the presence of 10 quaternary carbons with a carbon–oxygen bond, six olefinic carbons, and five methylene carbons. The signal at δ_C_ 117.10 (C-1′) is an indication of an orthoester moiety formed between C9/C13/C14 at δ_C_ 80.50 (C-9), 83.92 (C-13), and 78.96 (C-14). There was an indication of an ester moiety at δ_C_ 165.54 (C-1″) and an epoxy ring found at C-6 and C-7 at δ_C_ 60.57 and 64.73, respectively. The effect of the α,β-unsaturated ketone resulted in the downfield chemical shift for C-19 at δ_C_ 9.90 (Table 1). The presence of three alcohol groups, δ_C_ 64.07 (C-20), a primary alcohol; δ_C_ 72.17 (C-5), a secondary alcohol; and δ_C_ 71.98 (C-4), a tertiary alcohol, were detected. The HSQC and COSY correlation led to the assignment of the proton and carbon signals. The ^1^H and ^13^C chemical shifts are shown in Appendix A, respectively (Appendix A), where comparisons are drawn with the published literature [26]. Based on these results, the compound was confirmed to be yuanhuacine A (Figure 3A, compound **1**).

Compound **2** (as part of an isomeric mixture): The compound mixture was obtained as two isomers where yuanhuacine was identified as the one compound occurring within the isomeric mixture. The compound mixture was obtained as a pale-yellow amorphous powder. Based on UPLC-QTOF-MS/MS analysis, under ESI positive ionisation mode, yuanhuacine appeared at *m/z* 649.3761 [M+H]^+^ with a corresponding molecular formula of C_37_H_45_O_10_ (calculated for 649.3267), corresponding to that observed in literature for yuanhuacine [27]. The ^1^H-NMR spectra of the compound was found to be similar to compound **1** with additional protons at positions H-8′ (2H, δ_H_ 1.27) and H-9′ (2H, δ_H_ 1.31) (Table 1, Figure 3A, compound **2**) [26]. Based on the ^1^H-NMR spectra, the trans-C-2′/C-3′ of yuanhuacine can clearly be distinguished based on the large coupling constant between H-2′ (δ_H_ 5.67) and H-3′ (δ_H_ 6.70) (H-3′, *J* = 15.5, 10.58 Hz). Similarly, the trans-C-4′/C-5′ geometry is observed due to the splitting pattern (dd) and large coupling between H-4′ (δ_H_ 6.06) and H-5′ (δ_H_ 5.87) (H-4′, *J* = 15.59, 10.85 Hz). This observation corresponds well with the literature [26].

Interestingly, based on the smaller coupling constant observed between H-2′ (δ_H_ 6.58) and H-3′ (1H, t, *J* = 11.26 Hz, H-2′), we predict a unique cis-trans isomer, with the trans-C-4′/C-5′ geometry observed between H-4′ (δ_H_ 6.12) and H-5′ (δ_H_ 5.87) (H-4′, *J* = 14.83, 7.06 Hz); Figure 3B. However, due to the difficulty in purifying the mixture, it’s absolute structure could not be determined.

Compound **3**: The compound was identified as yuanhuajine and was obtained as a white amorphous powder. Based on the UPLC-QTOF-MS/MS analysis, under ESI positive ionisation mode, yuanhuajine was observed at *m/z* 647.2858 [M+H]^+^ with a corresponding molecular formula of C_37_H_42_O_10_ (calculated for 647.2856) with 17 degrees of unsaturation. The ^1^H-NMR spectrum revealed proton signals and ^13^C atoms in accordance with the molecular formula. The ^1^H- NMR spectrum revealed the presence of four methyl groups at resonance signal δ_H_ 1.88 (3H, s, H-17), 1.75 (3H, s, H-19), 1.41 (3H, t, *J* = 3.9 Hz, H-18), and 0.92 (3H, t, *J* = 7.4, H-10′). The spectrum also showed an aromatic system revealed by signals at δ_H_ 7.72 (2H, m, H-3′ and H-7′), 7.59 (1H, m, H-5′), and 7.39 (2H, m, H-4′and H-6′). The presence of a conjugated system was demonstrated by signals at δ_H_ 5.81 (1H, d, *J* = 15.9 Hz, H-5′), 7.39 (1H, m, H-2′), 7.39 (1H, m, H-3′), 6.20 (1H, m, H-4′), 6.6 (1H, m, C-5′), 6.16 (1H,m, H-6′), and 5.97 (1H, m, H-7′). The ^13^C NMR spectra revealed the presence of an ortho-ester moiety at resonance signals δ_C_ 119.3 (C-1′) joined from C9/13/14. The ^13^C signal at δ_C_ 209.50 (C-3) confirmed the presence of an α,β unsaturated ketone. Signals at δ_C_ 135.6 (C-2′), 126.3 (C-3′and 7′), and 128.2 (C-4′ and C-6′ and 129.9 (C-5′) confirmed the presence of an aromatic system and the aliphatic conjugated system was confirmed by signals at δ_C_ 135.3 (C-2′), 128.2 (C-3′), 127.2 (C-4′), δ 129.9 (C-5′), 127.2 (C-6′), and 141.5 (C-7′) (Table 1). The compound was confirmed as yuanhuajine (Figure 3A, compound **3**) by comparison with published data. Yuanhuajine and yuanhuacine A have similar basic backbones as their ^1^H-NMR signals are almost identical except for an extra resonance form seen for yuanhuajine and observed at δ_H_ 6.16 (H-6′) and δ_H_ 5.97 (H-7′) corresponding to δ_C_ 129.9 (C-6′) and δ_C_ 141.5 (C-7′), respectively.

Compound **4** was identified as gniditrin and was obtained as a white amorphous powder. Based on the UPLC-QTOF-MS/MS analysis, under ESI positive ionisation mode, gniditrin was observed at *m/z* 647.2858 [M+H]^+^ corresponding to molecular formula C_37_H_42_O_10_ (calculated, 647.2856) with 17 degrees of unsaturation. The ^1^H NMR of compound **4** was compared favourably with the published data of gniditrin, which confirmed its structure (Table 1, Figure 3A, compound **4**) [28].

### 3.4. Anti-HIV Activity of Isolated Compounds

Compounds **1**–**4** (with compound **2** occurring as an isomeric mixture) were tested next for anti-HIV replication activity in vitro using the deCIPhR assay in dose-response studies (Figure 4). Yuanhuacine A (**1**) consistently showed a pronounced activity by inhibiting HIV replication (ranging between 65–80%) across all test concentrations (ranging from 0.08 to 5 μg/mL). The cytotoxicity was assessed using Alamar Blue reagent under the same cell culture conditions used for the anti-HIV replication activity. After four days of incubation, there was no evidence of cytotoxicity at any concentration (data not shown). In contrast, yuanhuajine (**3**) exhibited a calculated EC_50_ of 1.8 μM (0.002 µg/mL), while gniditrin (**4**) was observed with an EC_50_ of 4.0 μM (0.006 µg/mL) and was the least active against HIV replication. All the isolated compounds belong to the genkwanine category of daphnane diterpenoids, with a 9,13,14 ortho-ester motif and a 6α-epoxy having been reported to be contributing factors to anti-HIV replication activity [25]. Gniditrin (**4**), with the weakest anti-HIV activity, has its phenyl group attached to position R_1_ and its alkyl (ester) chain linked at R_2_ [17,28,29]; in contrast, the remaining compounds have their alkyl chain linked in position R_1_ and the phenyl (ester) substituent linked in position R_2_ [30,31]_._ Both yuanhuacine A (**1**) and yuanhuacine (as an isomeric mixture) (**2**), with the strongest HIV inhibition, and have a double bond linked in their alkyl chain between carbons C-2′-C-3′ and C-5′-C-6′, while yuanhuajine (**3**) has an extra double bond between C-7′-C-8′ [25]. The absence of the double bond in yuanhuacine A (**1**) and yuanhuacine (as an isomeric mixture) (**2**) could therefore be important toward the anti-HIV activity of these compounds.

### 3.5. Anti-HIV Activity of Yuanhuacine A (1) in PBMC Isolates

To confirm the antiviral activity in the primary cell contact, yuanhuacine A (**1**) was re-assessed in PBMC using a subtype C viral strain (PBL286 (02ET_14)) as shown in Table 2. The nucleoside reverse transcriptase inhibitor AZT was used as a control with a measured EC_50_ of 0.007 µM and EC_90_ of 0.05 µM. Yuanhuacine A (**1**) demonstrated activity comparable to that of AZT, with an EC_50_ and EC_90_ of 0.03 and 0.09 µM, respectively. It further showed excellent selectivity over cytotoxicity with a selectivity index of > 500. The cytotoxicity of the compound was measured by staining the assay plates with tetrazolium-based dye MTS (CellTiter 96 Reagent, Promega, Madison, WI, USA).

### 3.6. Effects of Yuanhuacine A (1) and the Yuanhuacine-Containing Isomeric Mixture (2) on HIV Latency Reversal, Cell Viability, and T Cell Activation

To determine whether yuanhuacine A (**1**) and yuanhuacine (as an isomeric mixture) (**2**) also induced HIV latency reversal, we first assessed their activities in vitro using the J-Lat 10.6 cell line. These cells are derived from the Jurkat T cell line but contain a latently-infected, GFP-tagged HIV provirus [21]. The treatment of cells with LRAs for 24 h induces the production of viral proteins and GFP, which is monitored by flow cytometry [32]. Using this assay, we observed, for example, that the control PKC activator prostratin [33] induced a near-maximal GFP expression (i.e., latency reversal) in 66.0 ± 5.7% of the live-gated cells at 2.5 μM (Figure 5A). In contrast, both yuanhuacine A (**1**) and yuanhuacine (as an isomeric mixture) (**2**) exhibited similar levels of latency reversal (65.5 ± 6.3% and 58.2 ± 4.7%, respectively) at 0.15 μM, indicating ~16.7-fold enhanced activity over prostratin (Figure 5A).

When J-Lat 10.6 cells were treated in the presence of 1 μM of the pan-PKC inhibitor Gö 6983, we observed a 74.7 ± 17.3% reduction in latency reversal due to 2.5 μM prostratin (Figure 5B), consistent with prostratin′s function as a PKC activator. In contrast, the latency reversal induced by 0.1 μM Panobinostat, a control LRA that acts through histone deacetylase inhibition [14], was unaffected by Gö 6983, consistent with previous observations [32]. However, similar to prostratin, Gö 6983 inhibited 70.5 ± 12.0% and 71.5 ± 11.9% of latency reversal induced by 0.5 μM of yuanhuacine A (**1**) or yuanhuacine (as an isomeric mixture) (**2**), respectively (Figure 5B), indicating that both function as members of the PKC activator class of LRAs. Similar results were observed when the cells treated with prostratin, yuanhuacine A (**1**), or yuanhuacine (as an isomeric mixture) (**2**), but not panobinostat, and additionally treated with 2 μM enzastaurin, a selective PKC β isoform inhibitor (with53.7 ± 22.3%, 55.2 ± 25.6%, and 56.0 ± 27.0% inhibition of latency reversal, respectively), indicating that, similar to other diterpenes [34], yuanhuacine A (**1**) and yuanhuacine (as an isomeric mixture) (**2**) function in large part through the activation of these PKC isoforms.

To assess the effects of these compounds on cell viability, parental Jurkat cells lacking HIV provirus were treated with compounds for 24 h and then measured for cell viability using resazurin dye. Both Yuanhuacine A (**1**) and yuanhuacine (as an isomeric mixture) (**2**) affected cell viability in a manner similar to prostratin (Figure 5C). For example, the treatment of Jurkat cells with 0.5 μM prostratin resulted in an average 56.0 ± 10.0% viability (mean ± SD) relative to cells treated with 0.1% DMSO vehicle control, while treatment with 1.5 μM of Yuanhuacine A (**1**) or yuanhuacine (as an isomeric mixture) (**2**) resulted in 51.2 ± 11.2% and 58.5 ± 15.9% viability, respectively. These results indicate that the yuanhuacines induce more latency reversal at similar concentrations to prostratin without an additional loss of cell viability.

To determine the effects of the compounds on T cell activation, Jurkat cells were next incubated with compounds for 24 h and stained for the CD69 T cell activation marker. As expected, 2.5 μM prostratin induced an average 73.2 ± 1.3% (mean ± SD) CD69-positive cells (Figure 5D), consistent with prostratin’s role as an activator of PKC and T cells. In contrast, in cells treated with 0.1 μM Panobinostat, minimal CD69 expression was observed (1.5 ± 1.3% positive cells). However, when cells were treated with 0.5 μM of Yuanhuacine A (**1**) or yuanhuacine (as an isomeric mixture) (**2**), widespread CD69-positive cells were observed (75.4 ± 2.2% and 74.5 ± 1.3%, respectively; Figure 5D), consistent with these compounds also inducing T cell activation as PKC activators.

To establish HIV latency reversal in a primary cell context, we next obtained primary CD4+ cells directly from three HIV-infected, cART-suppressed donors (Figure 6). Briefly, 2 × 10^6^ CD4+ T-cells per donor were cultured in the presence of 0.1% DMSO, anti-CD3/CD28 dynabeads (a positive control CD4+ T cell activator), 10 μM prostratin, or 1.5 μM of yuanhuacine A (**1**) or yuanhuacine (as an isomeric mixture) (**2**). Following 24 h incubation, the cell viability was assessed by trypan blue staining, and the absolute viral RNA in each culture supernatant was determined by qRT-PCR [22]. Consistent with our previous observations, we observed considerable donor-to-donor variability [22]. Despite this, both yuanhuacine A (**1**) and yuanhuacine (as an isomeric mixture) (**2**) were generally well tolerated by CD4+ T cells as measured by direct cell counting with trypan blue (average 96.4 ± 32.2 and 79.6 ± 41.5% viability, respectively, compared to 65.9 ± 12.4% viability in the presence of prostratin; Figure 6A).

When assessed for total copies of supernatant viral RNA, we observed a baseline average of 32.6 ± 43.3 viral copies/mL across three donors, with one donor lacking viral RNA above the limit of detection in this assay (10 copies/mL; Figure 6B, dotted line). In contrast, anti-CD3/CD28 increased viral RNA expression from all three donors for an average of 8.8-fold over the unstimulated cells (average 287 ± 367 copies/mL). For the cells treated with 10 μM prostratin, we observed increased viral RNA across all donor samples for an average 23.2-fold increase (average 755.5 ± 1144.2 copies/mL). Similarly, 1.5 μM of yuanhuacine A (**1**) or yuanhuacine (as an isomeric mixture) (**2**) induced comparable increases in viral RNA to anti-CD3/CD28 that approached the activity of prostratin, despite a much lower concentration; we observed an average 10.8-fold increase in viral RNA for yuanhuacine A (**1**) (average 352.5 ± 430.4 copies/mL) and 9.1-fold increase for yuanhuacine (as an isomeric mixture) (**2**) (average 297 ± 267 copies/mL; Figure 6B). Taken together, these results indicate that both yuanhuacine A (**1**) and yuanhuacine (as an isomeric mixture) (**2**) are potent activators of HIV latency reversal ex vivo in addition to being potent inhibitors of viral replication.

## 4. Conclusions

The HIV inhibitory effect and latency reversal activity of daphnane diterpenoids have been reported previously for compounds isolated from other *Gnidia* species [35]. However, we show here, for the first time, that these compounds are also present in *G. sericocephala* roots, a readily-available plant traditionally used for HIV/AIDS management in South Africa. These small molecules also inhibit both viral replication and induce latency reversal similarly to prostratin and bryostatin, thereby prioritizing this plant as a source of these compounds for further study. Moreover, the latency reversal activity of yuanhuacine A (**1**) and yuanhuacine with its isomer (**2**) are reported for the first time here and support their use as prototypes toward the development of dual HIV antivirals with latency-reversing activity. The isolated compounds in this study share some structural similarities with known daphnane diterpenoids (gnidimacrin) and some phorbol esters (prostratin and bryostatin) that are reported to possess both anti-HIV replication activity, which acts at least in part through the down regulation of HIV CD4 and co-receptors (CCR5 and CXCR4) used by the virus in cell entry, as well as latency reversal activity, which acts through PKC activation and subsequent NF-κB signaling to drive proviral transcription [34,36]. Our results for yuanhuacine A (**1**) and yuanhuacine with its isomer (**2**), which show the inhibition of HIV replication and a requirement for PKC function for latency reversal in vitro, are consistent with these previous reports. PKC activators are not extensively assessed to date in the HIV shock and kill therapy. As a medicinal plant with latency-reversal properties acting through the PKC pathway, *G. sericocephala* could be a useful addition toward an eventual HIV eradication therapy. Moreover, the simultaneous blocking of HIV replication and breaking of latency shown by these compounds justifies further investigation on the mechanisms of this plant in additional cell and animal models. As South Africa continues to harbour a high number of people living with HIV that are resistant to available antiretroviral therapies and/or have limited access to existing cART, *G. sericocephala* root extracts and/or isolated compounds, as supported by both the reported traditional use and the reverse pharmacology-based studies described herein, are useful leads toward additional therapies for HIV/AIDS management to supplement cART in South Africa and other LMICs.

## Figures and Tables

**Figure 1 viruses-14-01437-f001:**
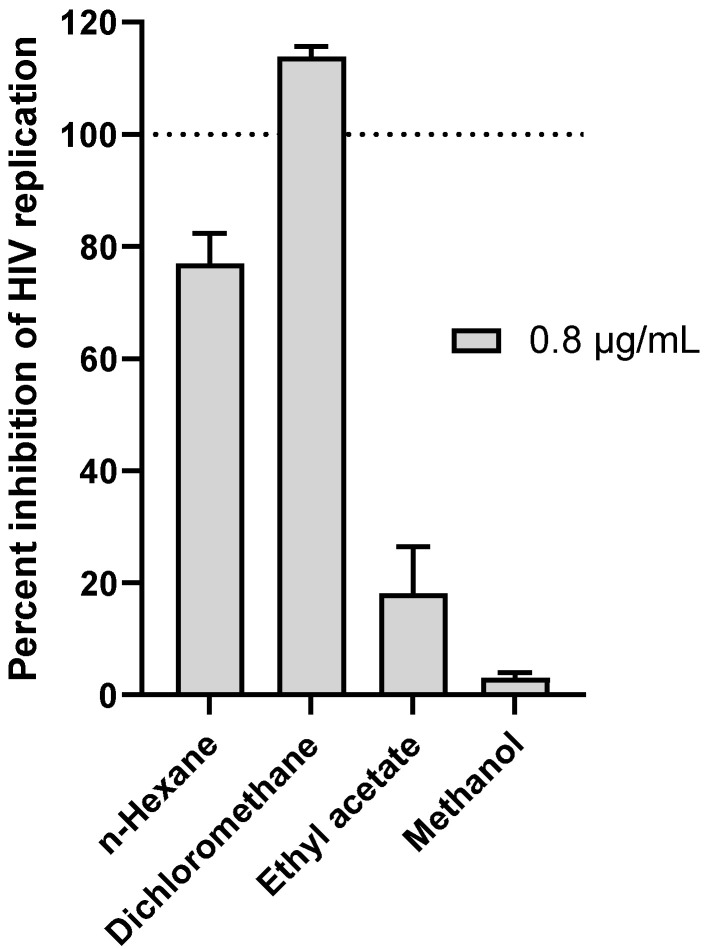
Anti-HIV replication activity of *G. sericocephala* root extracts at 0.8 μg/mL as measured using the in vitro *deCIPhR* assay. Briefly, the HeLa-SXR5 cell line was infected with the CXCR4-tropic HIV isolate NL4-3 [24], and infection was monitored in SXR5 cells due to β-galactosidase expression of an HIV long terminal repeat (LTR)-driven LacZ gene. Reduced or eliminated β-galactosidase activity in infected cells in the presence of test agent thus indicated inhibition of HIV replication.

**Figure 2 viruses-14-01437-f002:**
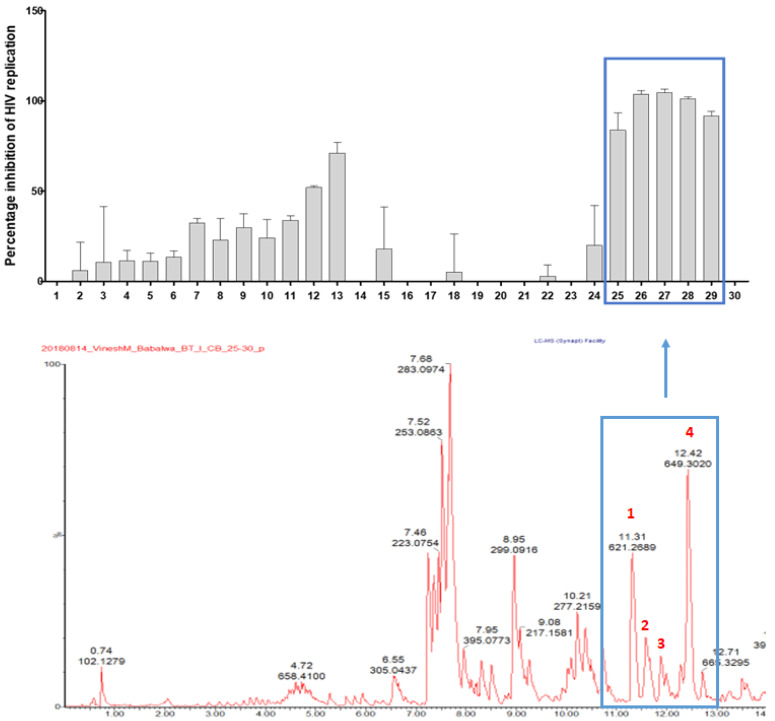
Anti-HIV replication activity of microfractions at 2.5 µg/mL, as measured using the deCIPhR in vitro assay, with corresponding UPLC-QTOF-MS data of the combined microfractions. Peaks 1–4 corresponded to isolated daphnane diterpenoids.

**Figure 3 viruses-14-01437-f003:**
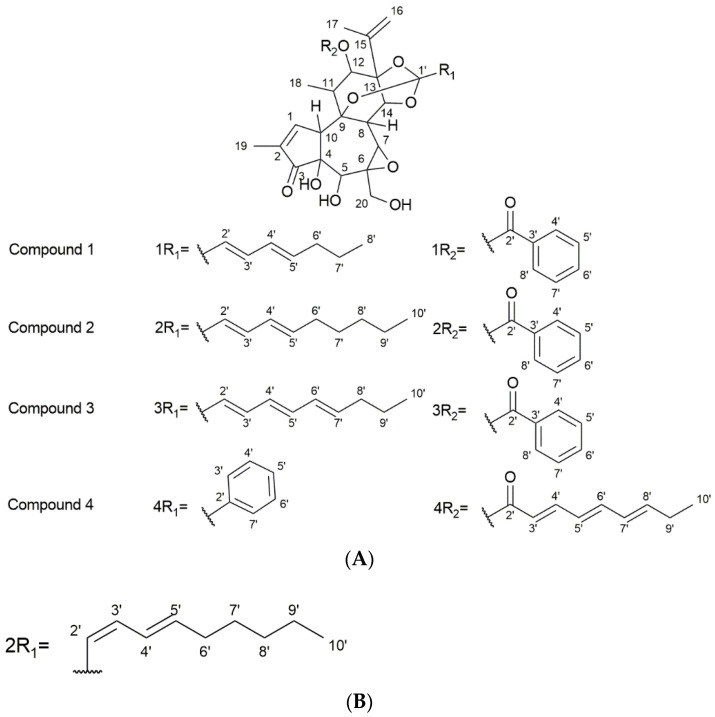
(**A**) Structures of the isolated compounds. (**B**) Proposed R_1_ fragment of the second compound present with compound **2** (isomer present as part of Yuanhuacine mixture).

**Figure 4 viruses-14-01437-f004:**
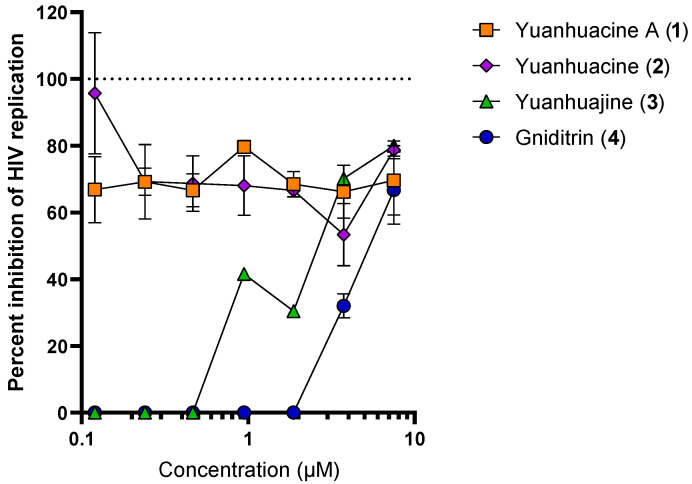
Anti-HIV replication activity of isolated compounds as measured using the deCIPhR in vitro assay.

**Figure 5 viruses-14-01437-f005:**
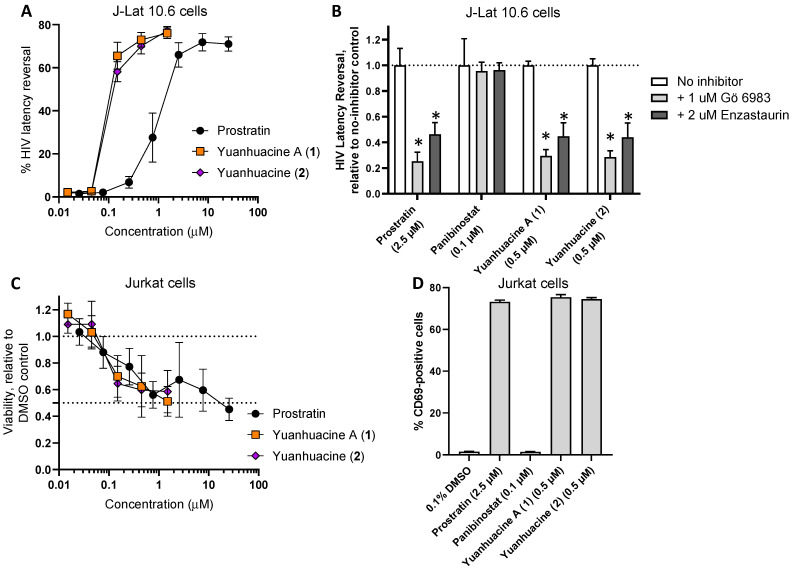
(**A**) Effects of compounds on HIV provirus expression in J-Lat 10.6 T cells following 24 h incubation, as measured by GFP reporter expression by flow cytometry. (**B**) Effects of pan-PKC inhibitor Gö 6983 (1 μM) and PKC β inhibitor enzastaurin (2 μM) on LRA-induced latency reversal. (**C**) Effects of compounds on cell viability in Jurkat T cells after 24 h, as measured by resazurin dye. (**D**) Effects of compounds on Jurkat T cell activation, as measured by CD69 expression, following 24 h incubation with compounds. Results denote the mean ± SD of at least 3 independent experiments. *, *p* < 0.01 by Student’s two-sided t test.

**Figure 6 viruses-14-01437-f006:**
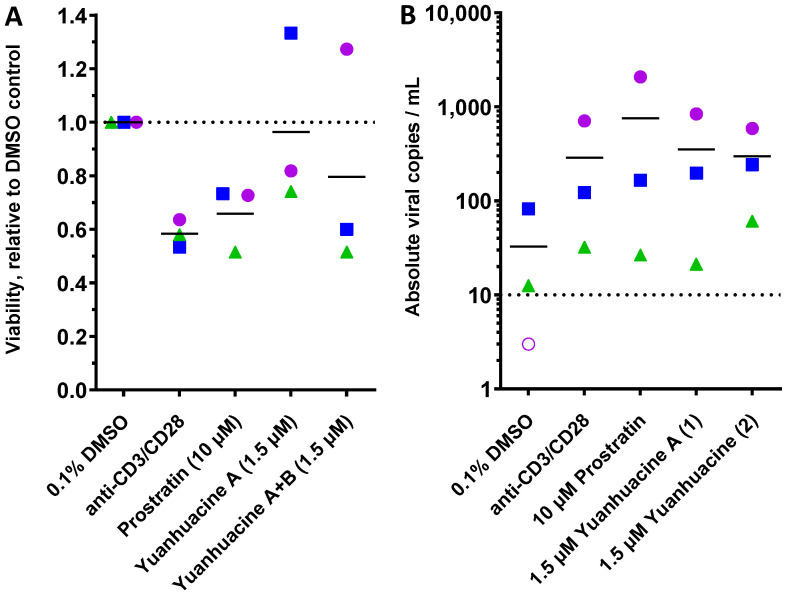
Effects of 10 μM prostratin and 1.5 μM of yuanhuacine A (**1**) or yuanhuacine (as an isomeric mixture) (**2**) on cell viability (**A**) and absolute viral RNA copies/mL of supernatant (**B**) in primary CD4+ cultures from 3 cART-suppressed donors with HIV following 24 h treatment. For both panels, shapes, and colors denote individual donors. In panel (**B**), the dotted line denotes limit of detection, and empty shape denotes no viral RNA detected.

**Table 1 viruses-14-01437-t001:** ^1^H and ^13^C NMR Spectroscopic Data for compounds **1**, **2**, **3** and **4** (CDCl_3_, 500 MHz).

Position	Compound						
	Yuanhuacine A (1)		Yuanhuacine * (2)		Yuanhuajine (3)		Gniditrin (4)
	δC	δH (*J*)	δC	δH (*J*)	δC	δH (*J*)	δH (*J*)
1	160.47, CH	7.48, t (8.06)	160.50, CH	7.62, s	160.05, CH	7.39, d (4.5)	7.59, s
2	136.96, C		137.04, C	-	136.98, C		
3	209.50, C		209.54, C	-	209.50, C		
4	71.98, C		60.58, C	-	78.37, C		
5	72.17, CH	4.25, s	64.07, CH	3.66, s	72.13, CH	4.27, s	4.27, s
6	60.57, C	-	72.15, C	-	60.54, C		
7	64.73, CH	3.84, d (13.0)	72.00, CH	4.27, s	64.41, CH	3.98, br d (7.6, 4.7)	3.48, s
8	34.80, CH	3.65, d (2.6)	35.7, CH	3.66, s	35.56, CH	3.81, dd (12.4, 12.7)	3.80, dt (5.7, 12.5)
9	80.50, C		78.40, C	-	72.22, C		
10	47.50, CH	3.87, quint (5.2)	47.3, CH	3.88, d (19.5)	47.51, CH	3.93, m	3.97, m
11	44.15, CH	2.58, q (7.3)	44.2, CH	2.57, m	44.21, CH	2.56, m	2.51, m
12	78.24, CH	5.24, s	78.9, CH	5.24, s	78.64, CH	5.11, t (7.9)	5.10, q (4.4, 8.5)
13	83.92, C		83.92, C	-	84.27, C		
14	78.96, CH	4.92, d (1.2)	81.2, CH	4.92, s	80.86, CH	4.93, d (2.5)	4.92, s
15	143.00, C		143.06, C	-	142.9, C		
16	113.67, CH_2_	5.02, m 5.03, s	114.62, CH_2_	5.04, t (8.4)	113.83, CH_2_	5.02, d (4.9)	5.02, d (4.12)
17	18.80, CH_3_	1.89, br s	18.79, CH_3_	1.90, s	18.37, CH_3_	1.88, s	1.88, s
18	18.37, CH_3_	1.41, d (7.69)	18.37, CH_3_	1.42, m	18.97, CH_3_	1.41, t (3.9)	1.41, t (3.9)
19	9.90, CH_3_	1.79, dd (1.27)	9.90, CH_3_	1.80, s	9.96, CH_3_	1.75, s	1.75, s
20	64.07, CH_2_	3.97, d (12.7), 3.67 s	64.72, CH_2_	3.97, d (19.5)	63.8, CH_2_	3.64, dd (7.6, 4.7)	3.63, dt (2.7, 3.1)
1′	117.10, C		117.1, C	-	119.3, C		
2′	122.29, CH	5.70, d (15.4)	122.41, CH	5.70, d (15.44)	135.3, CH	5.81, d (15.9)	
3′	135.24, CH	6.72, dd (10.87, 15.33)	135.4, CH	6.73, dd (11.07, 15.31)	128.2, CH	7.39, m	7.72, dd (4.9, 1.96)
4′	128.64, CH	6.09, dd (10.81, 15.3)	128.7, CH	6.08 (11.09, 15.03)	127.2, CH	6.20, m	7.39, d (6.2)
5′	139.20, CH	5.89, dt (15.3, 14.5)	139.6, CH	5.89, m (6.9,15.03)	129.9, CH	6.6, m	7.39, d (6.22)
6′	35.81, CH_2_	2.11, q (7.3)	32.8, CH_2_	2.12, d (7.35)	127.2, CH	6.16, m	7.39, d (6.22)
7′	22.24, CH_2_	1.44, sext (7.3)	28.9, CH_2_	1.41, dd (4.1, 6.7)	141.5, CH	5.97, m	7.59, s
8′	13.65, CH_3_	0.93, t (7.3)	31.4, CH_2_	1.27, s	36.1, CH_2_	2.1, q (8.7, 15.3)	
9′			22.68, CH_2_	1.31, br s	22.7, CH_2_	1.44, m	
10′			14.2, CH_3_	0.91, t (6.7)	13.8, CH_3_	0.92, t (7.4)	
1″	165.54, C		165.8, C	-	165.9, C		
2″	129.48, C		129.8, C	-	135.6, C		5.75, m
3″							7.39, d (6.22)
3″-7″	129.71, CH	7.92, dd (8.2, 8.4)	129.5, CH	7.92, d (7.6)	126.3, CH	7.72, m	
4″-6″	128.76, CH	7.48, t (8.1, 15.2)	128.8, CH	7.49, t (7.9)	128.2, CH	7.39, m	
4″							6.17, m
5″	133.45, CH	7.61, m	133.5, CH	7.62, s	129.9, CH	7.59, m	6.56, m
6″							5.95, m
7″							5.10, q
8″							2.1, q (8.7, 15.3)
9″							1.44, m
10″							0.92, t (7.4)

* Distinguishable from the isomeric mixture.

**Table 2 viruses-14-01437-t002:** Anti-HIV and selectivity of yuanhuacine A in PBMC.

Compound	Antiviral Activity (µM)		Cytotoxicity (µM)	Selectivity Index
	EC50	EC90	CC50	CC50/IC50
Yuanhuacine (1)	0.03	0.09	>15	>500
AZT	0.007	0.05	>1	>142

## Data Availability

Not applicable.

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
