# Peer review of "HPLC-Based Purification and Isolation of Potent Anti-HIV and Latency Reversing Daphnane Diterpenes from the Medicinal Plant Gnidia sericocephala (Thymelaeaceae)"

_viruses, 2022, doi:10.3390/v14071437_

Round 1
Reviewer 1 Report
The authors identify the need for novel agents to address HIV treatment (both viral suppression and reversal of HIV latency) and evaluate compounds present in a medicinal plant for these capabilities. The experiments are well designed and the data are clearly presented. Yuanhuacine A and yuanhuacine as part of a mixture both inhibit HIV as well as function as PKC agonists that could make them potentially useful as LRAs in LMIC. A common question that arises with these studies is how much of these compounds are present in the plant as it is ingested as part of traditional medicine? Are these trace compounds or do they represent a substantial portion of bioactive compounds in traditional medicine preparations of this plant? If the authors have any information about this, it would be helpful to include it.
The field of HIV persistence has a developed a tendency to be dismissive of PKC agonists due to concerns a narrow therapeutic window / likelihood of in vivo toxicity. The literature describing attempts to evaluate bryostatin as an anti-cancer agent provides much of the evidence that supports this opinion. One way to address this concern is through studies like this one - though taking this counter-argument a step further, it would be helpful to understand how much PKC agonists are being ingested in the whole plant as a traditional medicine preparation and whether these PKC agonists are the 'active' ingredients... if so, this would argue that PKC agonists are tolerated in vivo (and perhaps there are other naturally occurring compounds that offset some of the toxicity of PKC agonists?).
In the present study, it would be helpful to see the cell toxicity / viability data as well as CD69 expression with Yuanhuacine exposure (dose-response) alongside the anti-HIV and anti-latency results. CD69 is a helpful biomarker of PKC activation in this setting.
Author Response
Reviewer 1
The authors identify the need for novel agents to address HIV treatment (both viral suppression and reversal of HIV latency) and evaluate compounds present in a medicinal plant for these capabilities. The experiments are well designed and the data are clearly presented. Yuanhuacine A and yuanhuacine as part of a mixture both inhibit HIV as well as function as PKC agonists that could make them potentially useful as LRAs in LMIC. A common question that arises with these studies is how much of these compounds are present in the plant as it is ingested as part of traditional medicine? Are these trace compounds or do they represent a substantial portion of bioactive compounds in traditional medicine preparations of this plant? If the authors have any information about this, it would be helpful to include it.
Response: We thank the reviewer for this valuable comment. However, the quantitative analysis of the active daphnane diterpenoids in the DCM extract was not done. This is partly due to the fact that the DCM extract was produced for the purpose of isolation of the active compounds. The quantitative analysis of the compounds will be shown in a more suitable extract we produced for human consumption for an upcoming manuscript.
The field of HIV persistence has a developed a tendency to be dismissive of PKC agonists due to concerns a narrow therapeutic window / likelihood of in vivo toxicity. The literature describing attempts to evaluate bryostatin as an anti-cancer agent provides much of the evidence that supports this opinion. One way to address this concern is through studies like this one - though taking this counter-argument a step further, it would be helpful to understand how much PKC agonists are being ingested in the whole plant as a traditional medicine preparation and whether these PKC agonists are the 'active' ingredients... if so, this would argue that PKC agonists are tolerated in vivo (and perhaps there are other naturally occurring compounds that offset some of the toxicity of PKC agonists?).
In the present study, it would be helpful to see the cell toxicity / viability data as well as CD69 expression with Yuanhuacine exposure (dose-response) alongside the anti-HIV and anti-latency results. CD69 is a helpful biomarker of PKC activation in this setting.
Response: We appreciate this suggestion. Cell viability and CD69 expression data are now presented in Figures 5C and 5D, respectively.
Reviewer 2 Report
The study by Tembeni et al. expressed and purified anti-HIV agents from a medical plant and further demonstrated that the purified agents were able to inhibit HIV replication and reverse HIV latency. The overall manuscript was well-written and have adequate methods/results to demonstrate the effects of those two agents. Minor comments include
- The study described that Thymelaeaceae was selected based on its use by practitioners for HIV management in South Africa. I wonder if there’s any effectiveness or clinical benefits reported somewhere else?
- Figure 5: Are there any statistical differences calculated between groups?
- Line 254-258 describes experimental methods of HIV inhibition assay and should be included in the figure legends of figure 1.
Author Response
Reviewer 2
The study by Tembeni et al. expressed and purified anti-HIV agents from a medical plant and further demonstrated that the purified agents were able to inhibit HIV replication and reverse HIV latency. The overall manuscript was well-written and have adequate methods/results to demonstrate the
The study described that Thymelaeaceae was selected based on its use by practitioners for HIV management in South Africa. I wonder if there’s any effectiveness or clinical benefits reported some; where else?
Response: Non-peer reviewed data referring to clinical observational studies are now included in the supplementary information.
Figure 5: Are there any statistical differences calculated between groups?
Response: Statistical analyses for Figure 5B are now included.
Line 254-258 describes experimental methods of HIV inhibition assay and should be included in the figure legends of figure 1.
Response: We thank the reviewer for this valuable comment. The experimental methods of HIV inhibition assay have been included in Figure 1.
Reviewer 3 Report
Nowadays, the accessibility and drug resistance are still big challenges for HIV patients, especially in LMICs. In this study, the authors isolated two compounds from G. sericocephala. They showed that these compounds can both inhibit viral replication and induce latency reversal simultaneously, with minimum cytotoxicity, in in vitro and ex vivo models. Importantly, the sources of these compounds are readily available. Thus, though the further mechanism studies are warranted to better understand these compounds, they may represent use leads for additional HIV therapy in LMICs. The data in the study are solid and scientifically justified. However, I’d like to suggest several revisions. Specifically, NMR study played important role in this study. The author showed the chemical shifts in Table 1 and spectra in supporting material, which are useful. But the authors did not provide technique details for NMR experiment, both 1D proton and DEPT experiments, which are important information for this study. In addition, the legends of figures are too simple. Please add more details.
Author Response
Reviewer 3
Nowadays, the accessibility and drug resistance are still big challenges for HIV patients, especially in LMICs. In this study, the authors isolated two compounds from G. sericocephala. They showed that these compounds can both inhibit viral replication and induce latency reversal simultaneously, with minimum cytotoxicity, in in vitro and ex vivo models. Importantly, the sources of these compounds are readily available. Thus, though the further mechanism studies are warranted to better understand these compounds, they may represent use leads for additional HIV therapy in LMICs. The data in the study are solid and scientifically justified. However, I’d like to suggest several revisions. Specifically, NMR study played important role in this study. The author showed the chemical shifts in Table 1 and spectra in supporting material, which are useful. But the authors did not provide technique details for NMR experiment, both 1D proton and DEPT experiments, which are important information for this study.
Response: We thank the reviewer for raising this point. A subsection on NMR experimental method in line 191-195 as shown below
The 1H, 13C and 2D NMR spectral data were acquired on a Bruker Avance III HD 500 MHz NMR spectrophotometer at 25°C with Prodigy Probe. The compounds were dissolved in deuterated chloroform (CDCl3) (Aldrich Chemistry, Sigma-Aldrich, USA) and the Chemical shifts reported in ppm, referenced to residual solvent resonances (CDCl3 δH 7.26, δC77.16 ppm) (Pierens, 2014). The citation was also included in the bibliography as reference (27).
Detailed description of the NMR analysis is not provided as these are known compounds and the NMR data is compared to that of the published data with references provided.
In addition, the legends of figures are too simple. Please add more details.
Response: Additionally, the figures in the Supplementary data have been updated with more details with an additional figure showing the DEPT spectra (Figure S3, supplementary data).
Reviewer 4 Report
Comment 1: In Figure 5. B, the size of labels on X-axis should match,
i.e., 'Yuanhuacine A (1)' and 'Yuanhuacine (2)'.
Comment 2: Please, describe the pathway related with anti-HIV effect of Gnidia sericocephala , components 1, and components 2 In Discussion.
Author Response
Reviewer 4
Comment 1: In Figure 5. B, the size of labels on X-axis should match,
i.e., 'Yuanhuacine A (1)' and 'Yuanhuacine (2)'.
Response: Corrected
Comment 2: Please, describe the pathway related with anti-HIV effect of Gnidia sericocephala , components 1, and components 2 In Discussion.
Response 2: We thank the reviewer for this valuable comment. The pathway related with Gnidia sericocephala, compound (1) and compound (2) has been added in line 299-301.